# Effective Elastic Modulus of Wavy Single-Wall Carbon Nanotubes

Chensong Dong

School of Civil and Mechanical Engineering, Curtin University, GPO Box 1987,
Perth 6845, Western Australia, Australia; c.dong@curtin.edu.au

**Abstract:** A simple method for determining the effective elastic modulus of wavy single-wall carbon nanotubes (SWCNTs) is presented in this paper. The effective modulus of curved SWCNTs is derived using Castigliano's theorem. The effect of curvature on the effective modulus is studied. This method is verified by finite element analysis (FEA). The distributions of effective moduli are studied by Monte Carlo simulation. The effective modulus of a general wavy SWCNT is derived by considering the SWCNT as a number of curved SWCNT sections.

**Keywords:** carbon nanotube (CNT); modulus; waviness; Monte Carlo simulation

## 1. Introduction

Since the discovery of multi-wall carbon nanotubes (MWCNTs) in 1991 by Iijima [1], and subsequent synthesis of single-wall carbon nanotubes (SWCNTs) [2,3], numerous experimental and theoretical studies have been carried out to investigate the electronic, chemical, and mechanical properties of CNTs. SWNT–polymer composites are theoretically predicted to have both exceptional mechanical and special functional properties that carbon fibre–polymer composites cannot offer [4].

Previous studies on the mechanical properties of nanotubes utilized computational methods, such as molecular dynamics and ab initio models. In general, these computational studies have found nominal values for the axial Young's modulus on the order of 1 TPa [1,2]. Deng et al. [5] found the effective modulus is in the range 530–700 GPa from Raman characterisation of highly aligned electrospun SWNT/PVA nanocomposite fibres and SWNT/PVA nanocomposite films.

It has been found that CNTs are highly wavy when dispersed in a polymer. Waviness refers to the degree of bending or curvature present in the structure of individual nanotubes. Carbon nanotubes can exhibit different degrees of waviness depending on their synthesis method, growth conditions, and structural defects. The waviness of CNTs is influenced by several factors, including the diameter, length, chirality (structural arrangement of carbon atoms), and the presence of impurities or structural defects.

In general, CNTs can have varying degrees of waviness. Some nanotubes are relatively straight and have minimal curvature, while others exhibit more pronounced bending or twisting. The waviness of CNTs can be influenced by external forces, such as mechanical stress or interactions with neighbouring nanotubes or other materials.

In general, waviness in CNTs can have a detrimental effect on their properties and make them less useful for certain applications [6]. Waviness in CNTs can alter their electronic structure, leading to changes in their electronic transport properties, such as their conductivity and density of states. This can result in a reduction of the intrinsic carrier mobility and an increase in the resistance [7,8]. Waviness can also affect the mechanical properties of CNTs, leading to decreased stiffness and strength. This can result in a reduced ability of CNTs to withstand loads and deformations.

When being used to reinforce a polymer, the waviness of CNTs can limit the modulus enhancement of the composite, resulting in improvements that are less than predicted by traditional theories. Thus, waviness needs to be considered when studying the mechanical

properties of CNTs. Fisher et al. [9] and Bradshaw et al. [10] extended the Mori–Tanaka method to composites consisting of more than one type of inclusion type via the method introduced by Tendon and Weng [11] so that the distribution of CNT waviness can be considered. They also demonstrated the effect of CNT waviness distribution by introducing a discrete probability distribution. Maghsoudlou et al. [12] demonstrated the necessity of consideration of three important parameters, including SWCNTs' interphase, curvature, and agglomerations. Ginga et al. [13] demonstrated that the waviness of the CNT played a critical role in the significant increase of the compliance values (very low effective modulus) by orders of magnitude. Shi et al. [6] found that waviness and agglomeration might reduce the stiffening effect of nanotubes significantly. Shao et al. [14] studied the effects of the waviness of the CNTs and the interfacial debonding between them and the matrix on the effective moduli of CNT-reinforced composites. A simple analytical model was presented to investigate the influence of the waviness on the effective moduli. Zhu and Narh [15] utilized numerical simulation to study the impact of nanotube curvature on the tensile modulus of polymer composites reinforced with carbon nanotubes. Results from their simulations indicated that the tensile modulus of a nanotube-reinforced composite undergoes a significant decrease when nanotubes deviate from their axial direction. Moreover, the presence of curved nanotubes in the polymer matrix has a substantial impact on reducing the composite modulus.

The curvature of SWCNTs also has a significant effect on buckling. Shima [16] presented a review on the buckling of CNTs. Li et al. [17] studied the nonlinear in-plane instability of shallow circular arches made of functionally graded carbon-nanotube-reinforced composite that were limited by rotational symmetry and put under constant radial stress in a temperature environment.

In this paper, a simple method based on Castigliano's theorem for determining the effective elastic modulus of SWCNTs is presented. This method is verified by finite element analysis (FEA). Using this method, the effect of curvature on the effective modulus is studied. The statistical distributions of the effective moduli are studied by Monte Carlo simulation. A general wavy SWCNT is considered as a number of curved SWCNT sections being connected in series. The effective modulus for each section can be derived using the developed approach, and the effective modulus of the general wavy SWCNT can be derived accordingly.

## 2. Approach

### 2.1. Analytical Model

A simple analytical model for predicting the effective modulus of SWCNTs is developed based on Castigliano's theorem. For simplicity, a curved SWCNT is considered, as schematically shown in Figure 1. The SWCNT is subjected to point forces, $P$, at both ends. The diameter of SWCNTs varies. Chen et al. [18] presented a method to both precisely and continuously control the average diameter of single-walled carbon nanotubes in a forest ranging from 1.3 to 3.0 nm with 1 A° resolution. Fagan et al. [19] demonstrated the effective separation of single-wall carbon nanotube (SWCNT) species with diameters 1.3–1.6 nm through multistage aqueous two-phase extraction (ATPE), including isolation at the near-monochiral species level up to at least the diameter range of SWCNTs synthesized by electric arc synthesis. Charlier and Lambin [20] obtained SWCNTs of the diameter 1.3 nm. Park et al. [21] synthesised SWCNTs with diameter distributions peaked at 0.9 and 1.3 nm. Shi et al. [22] found that the SWCNTs had nearly the same diameter of 1.3 nm and belonged to armchair $(n, n)$-type carbon nanotubes ($n$ = 8, 9, 10 and 11). Jinno et al. [23] also found that the diameter of SWCNTs was approximately 1.3 nm. In this study, the outer diameter and wall thickness of the SWCNT are taken to be 1.3 nm and 0.142 nm, respectively [18–26]. The curve is characterised by an arc. Given the chord length, $L$, and the arc height, $h$, the radius of the arc, $R$, is given by

$$R = \frac{L^2}{8h} + \frac{h}{2} = \frac{L}{8\lambda} + \frac{L\lambda}{2} \tag{1}$$

where $\lambda = \frac{h}{L}$.

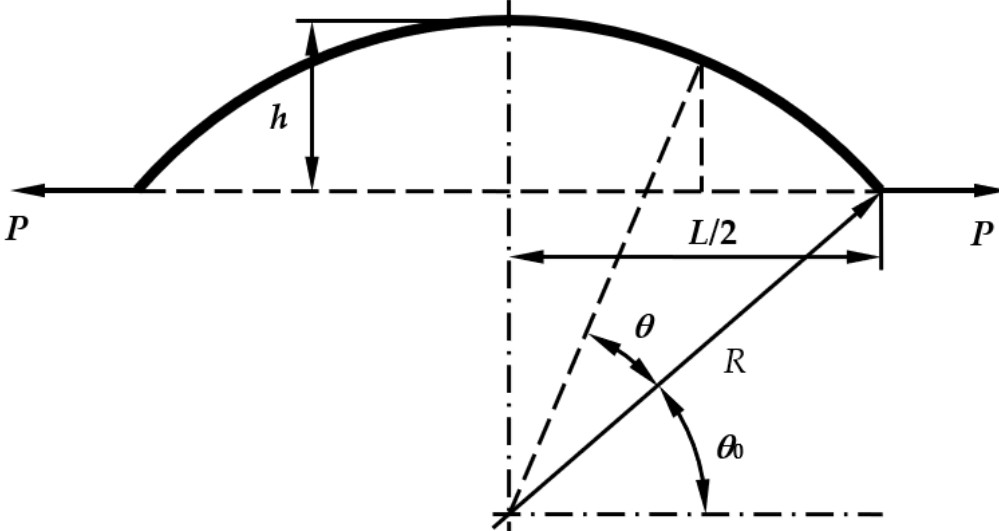

**Figure 1.** A curved SWCNT.

The angle $\theta_0$ is given by

$$\theta_0 = \cos^{-1}\left(\frac{L}{2R}\right) = \cos^{-1}\left(\frac{4Lh}{L^2 + 4h^2}\right) = \cos^{-1}\left(\frac{4\lambda}{1 + 4\lambda^2}\right) \tag{2}$$

Applying Castigliano's theorem, the displacement at one point is found by differentiating the strain energy, $U$, with respect to the load, $P$, acted on that point, i.e.,

$$\delta = \frac{dU}{dP} \tag{3}$$

The strain energy is due to bending, tension and shear, i.e.,

$$U = U_{bending} + U_{tension} + U_{shear} \tag{4}$$

Because of the symmetry, the strain energy due to bending is given by

$$U_{bending} = \frac{P^2 R^3}{E_{CNT}I}\int_0^{\pi/2 - \theta_0}[\sin(\theta + \theta_0) - \sin\theta_0]^2 d\theta \tag{5}$$

where $E_{CNT}$ and $I$ are the nominal elastic modulus and second moment of inertia, respectively.
The strain energy due to tension is given by

$$U_{tension} = \frac{P^2 R}{E_{CNT}A}\int_0^{\pi/2 - \theta_0}\sin^2(\theta + \theta_0)d\theta \tag{6}$$

where $A$ is the cross-sectional area.
The strain energy due to shear is given by

$$U_{shear} = \frac{P^2 R}{G_{CNT}A\gamma}\int_0^{\pi/2 - \theta_0}\cos^2(\theta + \theta_0)d\theta \tag{7}$$

where $G_{CNT}$ is the nominal shear modulus, and $\gamma$ is the Timoshenko shear coefficient (approximately 0.9).

The total displacement is given by

$$
\begin{aligned}
\delta &= \delta_{bending} + \delta_{tension} + \delta_{shear} \\
&= \frac{2PR^3}{E_{CNT}I}\left[\frac{1}{2}\left(\frac{\pi}{2}-\theta_0\right) - \frac{3}{4}\sin 2\theta_0 + \left(\frac{\pi}{2}-\theta_0\right)\sin^2\theta_0\right] \\
&+ \frac{2PR}{E_{CNT}A}\left(\frac{\pi}{4} - \frac{\theta_0}{2} + \frac{\sin 2\theta_0}{4}\right) + \frac{2PR}{G_{CNT}A\gamma}\left(\frac{\pi}{4} - \frac{\theta_0}{2} - \frac{\sin 2\theta_0}{4}\right)
\end{aligned}
\tag{8}
$$

The effective modulus is then given by

$$
E_{CNTeff} = \frac{PL}{\delta A} = \frac{E_{CNT}L}{2R\left\{\begin{array}{l} \frac{R^2 A}{I}\left[\frac{1}{2}\left(\frac{\pi}{2}-\theta_0\right) - \frac{3}{4}\sin 2\theta_0 + \left(\frac{\pi}{2}-\theta_0\right)\sin^2\theta_0\right] \\ + \left(\frac{\pi}{4} - \frac{\theta_0}{2} + \frac{\sin 2\theta_0}{4}\right) + \frac{2(1+\nu_f)}{\gamma}\left(\frac{\pi}{4} - \frac{\theta_0}{2} - \frac{\sin 2\theta_0}{4}\right) \end{array}\right\}}
\tag{9}
$$

where $\nu_f$ is the Poisson's ratio, which can be taken to be around 0.3.

### 2.2. Finite Element Analysis

The nanocomposite reinforced by a curved SWCNT is modelled by finite element analysis (FEA) for comparison purpose. Epoxy is chosen to be the matrix, and its modulus is 3.1 GPa. For simplicity, the nanocomposite is modelled as a rectangular prism, and the SWCNT is modelled as a curved hollow cylinder. Prefect bond is assumed between the SWCNT and the matrix. As an example, when $L = 50$ nm and $h = 1$ nm, the FEA model is shown in Figure 2. A 3D solid analysis is chosen, and a quarter of the composite is modelled by applying symmetry boundary conditions to the two planes of symmetry. The SWCNT volume fraction is determined from the volumes of the SWCNT and the matrix. An axial displacement 0.1 nm is applied to the end face. The reaction force due to this displacement is obtained from the FEA. The effective modulus of the composite is given by

$$
E_{eff} = \frac{FL}{\delta A}
\tag{10}
$$

where $F$ is the reaction force, $L$ is the half-length of the rectangular prism (because symmetry is applied), $\delta$ is the axial displacement, and $A$ is the cross-sectional area of the end face.

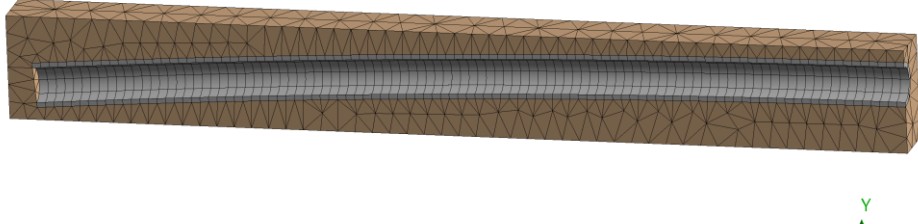

**Figure 2.** FEA model for SWCNT-reinforced composite.

The effective modulus of the SWCNT is determined with the rule of mixtures (RoM), given by

$$
E_{CNTeff} = \frac{E_{eff} - E_m(1 - V_{CNT})}{V_{CNT}}
\tag{11}
$$

where $E_{eff}$ is the effective modulus of the SWCNT-reinforced composite, $E_m$ is the modulus of the matrix, and $V_{CNT}$ is the SWCNT volume fraction.

### 2.3. Monte Carlo Simulation

The effective moduli of SWCNTs are statistically studied via Monte Carlo simulation using Equation (9). $h/L$ is assumed to be uniformly distributed, and the SWCNT length is assumed to be normally distributed. Different distributions are considered, as shown in

Table 1. In total, eight simulations are run to study the effect of the range of $h/L$ ratios, the average of the lengths of SWCNTs, and the standard deviation of the lengths of SWCNTs. In each simulation, for $h/L$ or $L$, random sampling is performed by generating 1000 random numbers based on the given distribution. The resulting effective moduli of SWCNTs are analysed statistically.

**Table 1.** Monte Carlo simulations.

| Set | Simulation | *h/L* | SWCNT Length (nm) |
|---|---|---|---|
| 1 | 1 | $U(0, 0.0025)$ | $N(500, 100)$ |
|   | 2 | $U(0, 0.005)$ | $N(500, 100)$ |
|   | 3 | $U(0, 0.01)$ | $N(500, 100)$ |
|   | 4 | $U(0, 0.02)$ | $N(500, 100)$ |
|   | 5 | $U(0, 0.01)$ | $N(1000, 100)$ |
|   | 6 | $U(0, 0.01)$ | $N(300, 100)$ |
|   | 7 | $U(0, 0.01)$ | $N(1000, 50)$ |
|   | 8 | $U(0, 0.01)$ | $N(1000, 200)$ |
| 2 | 1 | $N(0.005, 0.001)$ | $N(500, 100)$ |
|   | 2 | $N(0.01, 0.001)$ | $N(500, 100)$ |
|   | 3 | $N(0.02, 0.001)$ | $N(500, 100)$ |
|   | 4 | $N(0.01, 0.002)$ | $N(500, 100)$ |
|   | 5 | $N(0.01, 0.003)$ | $N(500, 100)$ |
|   | 6 | $N(0.01, 0.001)$ | $N(300, 100)$ |
|   | 7 | $N(0.01, 0.001)$ | $N(1000, 100)$ |
|   | 8 | $N(0.01, 0.001)$ | $N(1000, 50)$ |
|   | 9 | $N(0.01, 0.001)$ | $N(1000, 200)$ |
| 3 | 1 | $U(0, 0.0025)$ | $U(300, 700)$ |
|   | 2 | $U(0, 0.005)$ | $U(300, 700)$ |
|   | 3 | $U(0, 0.01)$ | $U(300, 700)$ |
|   | 4 | $U(0, 0.02)$ | $U(300, 700)$ |
|   | 5 | $U(0, 0.01)$ | $U(100, 500)$ |
|   | 6 | $U(0, 0.01)$ | $U(800, 1200)$ |

*2.4. Waviness*

A general wavy SWCNT can be decomposed into several curved SWCNT sections being connected in series. An example is given in Figure 3, which is decomposed into three curved SWCNT sections. For a general wavy SWCNT, its effective modulus is given by

$$\frac{1}{E_{CNTeff}} = \sum_{i=1}^{N} \frac{1}{E_{CNTeffi}} \tag{12}$$

where $E_{CNTeffi}$ is the effective modulus of the *i-th* SWCNT section, which can be determined from $L_i$ and $h_i/L_i$.

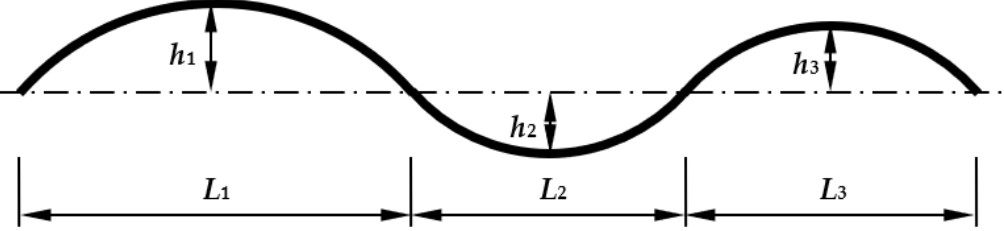

**Figure 3.** A general waved SWCNT.

## 3. Results

### 3.1. Effect of Curvature

The effect of curvature on the effective modulus is graphically shown in Figure 4, from which it is seen that the effective modulus decreases with both increasing $L$ and increasing $h/L$. Since a fixed SWCNT diameter is used in the simulation, longer SWCNTs have larger aspect ratios and thus are easier to be bent. Therefore, a small increase in $h/L$ results in a sharp decrease in the effective modulus.

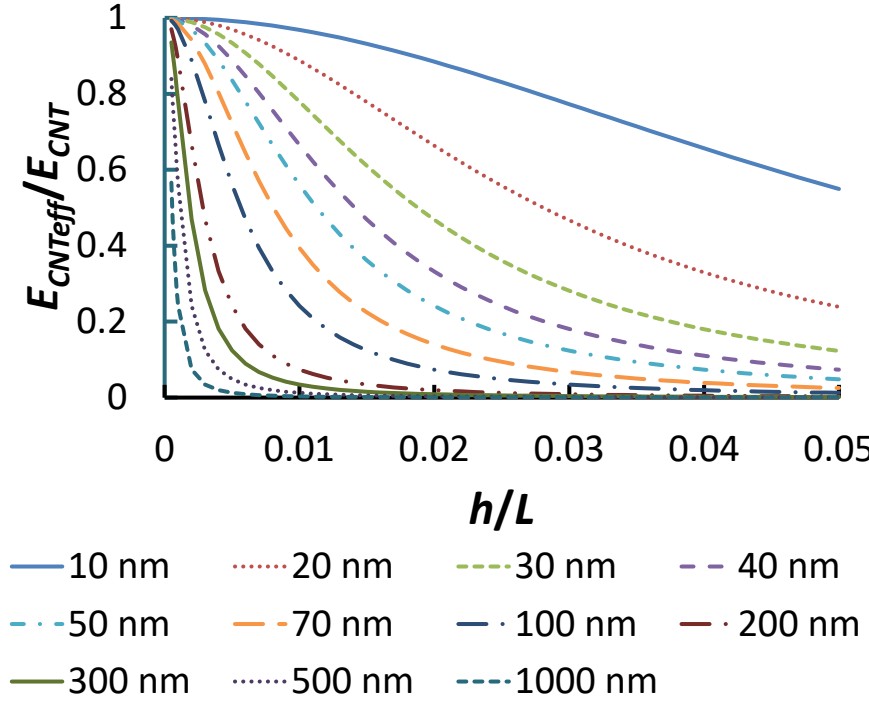

**Figure 4.** Effective modulus vs. $h/L$ for different SWCNT lengths.

### 3.2. Results from FEA

When $L$ = 50 nm and $h$ = 1 nm, the displacement in the $x$ direction from FEA is shown in Figure 5, in which the contours stand for the displacement in nm. The load is shown being transferred to the SWCNT near the end of the SWCNT. Figure 6 shows the stress distribution in the composite, in which the contours stand for the stress in MPa.

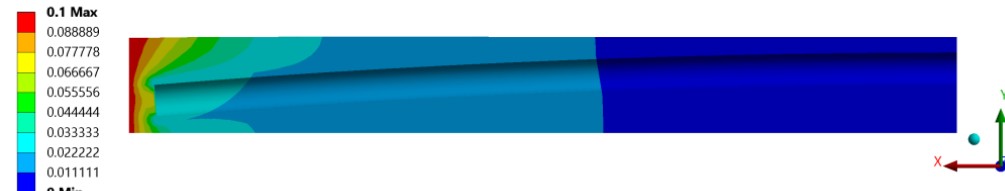

**Figure 5.** Displacements in SWCNT-reinforced composite.

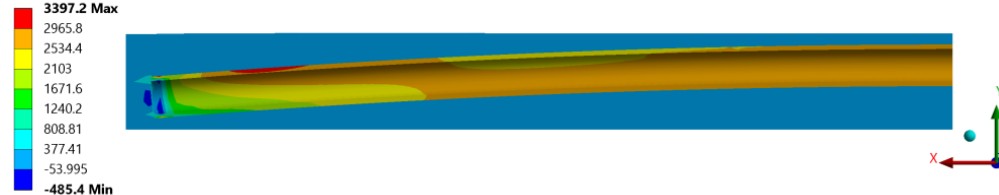

**Figure 6.** Stresses in SWCNT-reinforced composite.

For three different SWCNT lengths, the effective moduli from the analytical model and FEA are shown in Figure 7 and Table 2. It is seen that the analytical results and FEA results are in good agreement, with the relative difference being less than 10%. These results prove that the effective modulus of a curved SWCNT is significantly less than the nominal value.

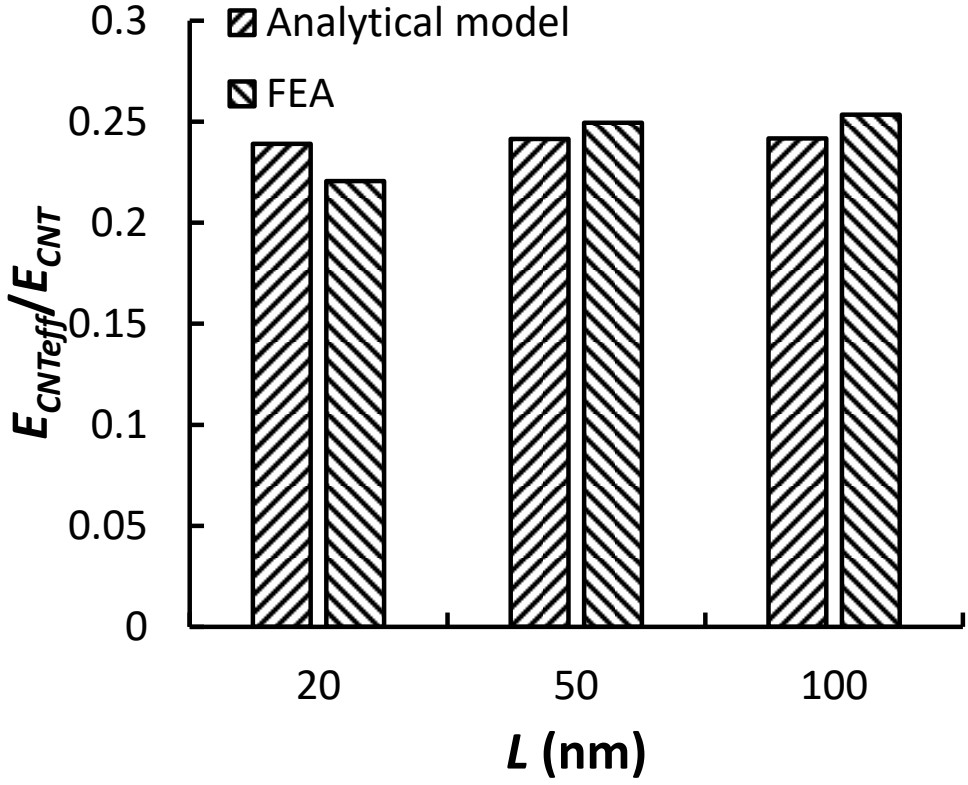

**Figure 7.** Effective moduli from analytical model and FEA.

**Table 2.** Effective moduli from analytical model and FEA.

| | Effective Modulus, $E_{CNTeff}/E_{CNT}$ | | |
| --- | --- | --- | --- |
| $L$ (nm) | Analytical Model | FEA | Relative Difference |
| 20 | 0.2390 | 0.2205 | 8.37% |
| 50 | 0.2414 | 0.2495 | −3.24% |
| 100 | 0.2417 | 0.2534 | −4.63% |

*3.3. Effect of Statistical Distributions*

When $h/L$ follows $U(0, 0.01)$ and $L$ follows $N(500, 100)$, the histogram of the effective moduli is shown in Figure 8, together with the cumulative percentage.

The data for the effective moduli are fitted into Weibull distributions. The shape parameters, scale parameters, means and medians from Monte Carlo simulations are given in Appendix A. When the SWCNT length follows $N(500, 100)$, and $h/L$ follows uniform distributions of different ranges, the fitted Weibull distributions are shown in Figure 9. It is noted that when $h/L$ follows $U(0, 0.0025)$, the maximum probability occurs at $E_{CNTeff}/E_{CNT}$ = 0.3478. As the range of $h/L$ increases, the maximum probability tends to occur when the effective modulus approaches zero. Figure 9 also shows the mean and median of the effective moduli vs. the range of $h/L$. When the range of $h/L$ increases, both the mean and median of the effective moduli decrease.

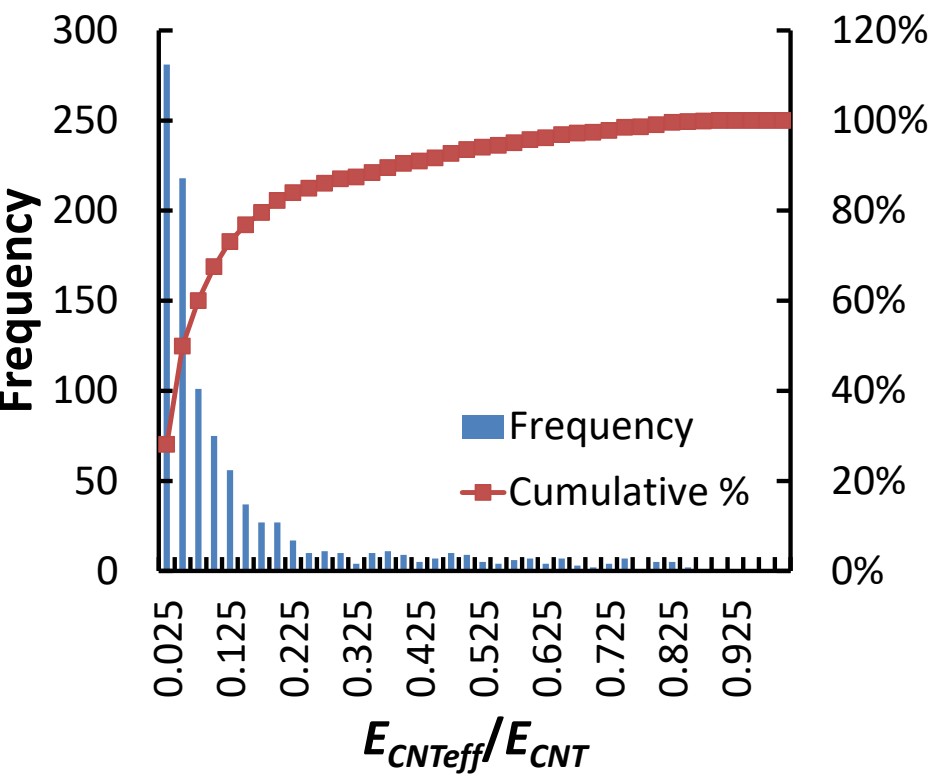

**Figure 8.** Histogram and cumulative percentage of effective moduli of SWCNTs when $h/L$ follows $U(0, 0.01)$ and $L$ follows $N(500, 100)$.

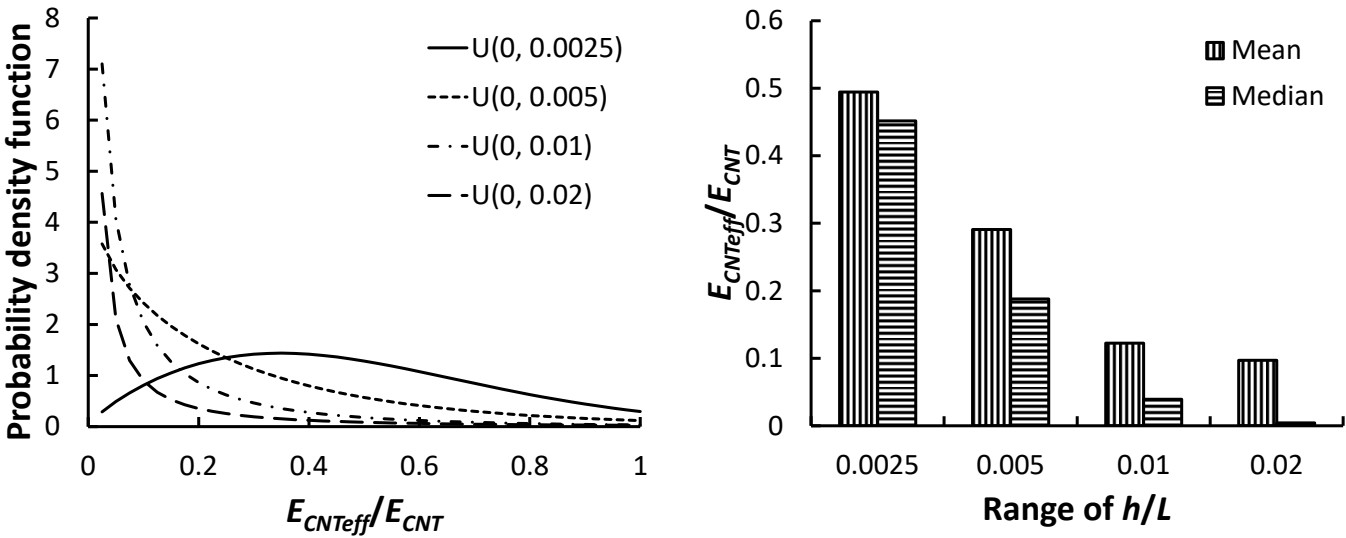

**Figure 9.** Effect of $h/L$ range on Weibull distribution for effective SWCNT moduli.

The fitted Weibull distributions are shown in Figure 10 for when $h/L$ follows $U(0, 0.01)$, and the SWCNT length follows normal distributions of different means and the same standard deviation, 100 nm. Figure 10 also shows the mean and median of the effective moduli vs. the mean of SWCNT lengths. It is noted that when the mean of the SWCNT lengths increases, both the mean and median of the effective moduli decrease.

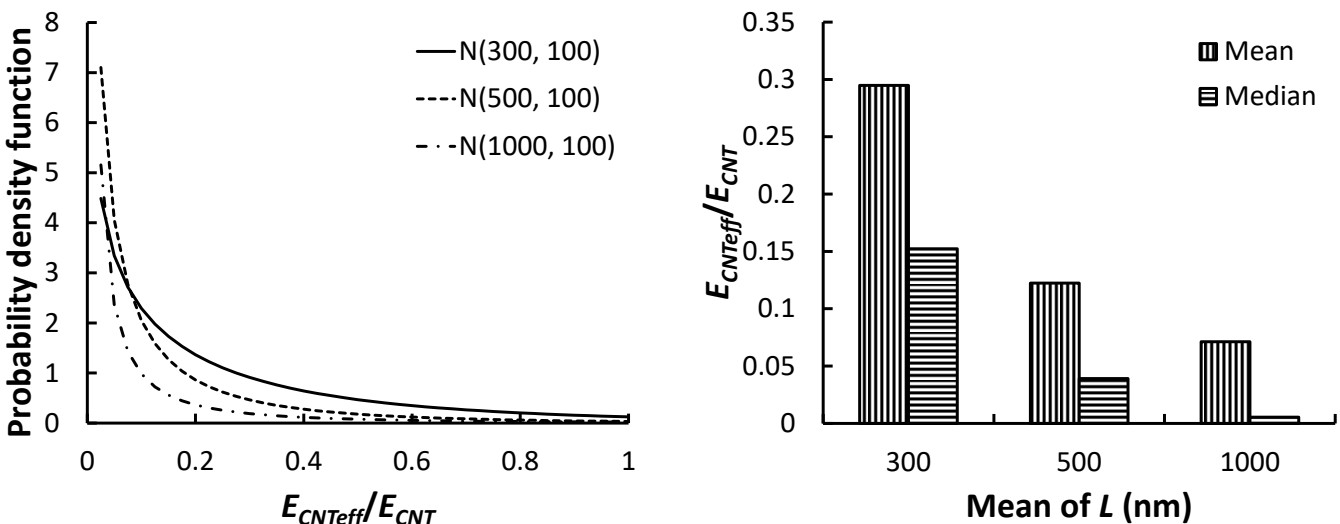

**Figure 10.** Effect of average SWCNT length on Weibull distribution for effective SWCNT moduli.

The fitted Weibull distributions are shown in Figure 11 for when $h/L$ follows $U(0, 0.01)$, and the SWCNT length follows normal distributions of different standard deviations and the same mean, 1000 nm. Figure 11 also shows the mean and median of the effective moduli vs. the standard deviation of SWCNT lengths. It is noted that the standard deviation of the SWCNT lengths has little effect on either the mean or the median of the effective moduli. Overall, all Weibull distributions skew to the right.

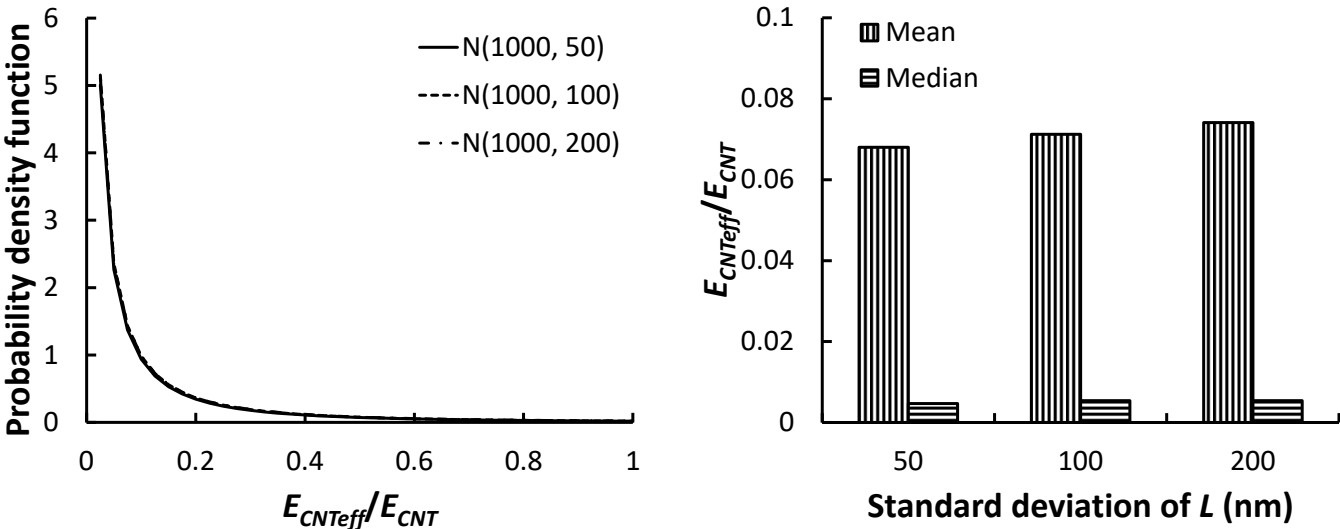

**Figure 11.** Effect of SWCNT length standard deviation on Weibull distribution for effective SWCNT moduli.

The fitted Weibull distributions are shown in Figure 12 for when the SWCNT length follows $N(500, 100)$ and $h/L$ follows normal distributions of different means. Figure 12 also shows the mean and median of the effective moduli vs. the range of $h/L$. When the mean of $h/L$ increases, both the mean and median of the effective moduli decrease.

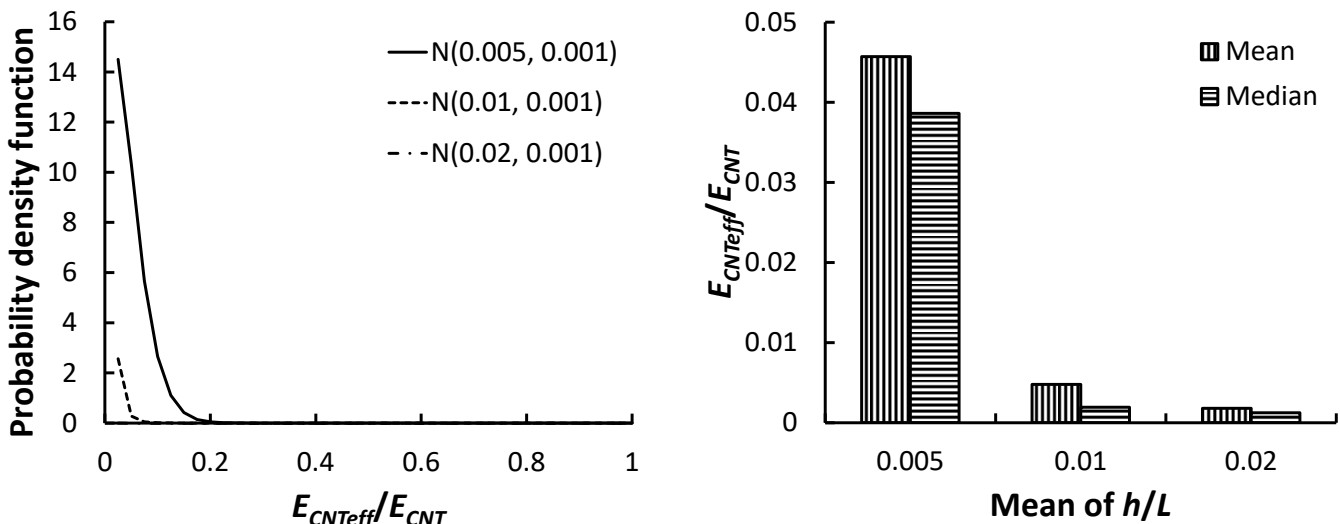

**Figure 12.** Effect of $h/L$ mean on Weibull distribution for effective SWCNT moduli.

The fitted Weibull distributions are shown in Figure 13 for when the SWCNT length follows $N(500, 100)$ and $h/L$ follows normal distributions of different standard deviations. Figure 13 also shows the mean and median of the effective moduli vs. the range of $h/L$. It is noted that the standard deviation of the SWCNT lengths has little effect on the median of the effective moduli. For the mean of the effective moduli, it is shown that the mean of the effective moduli increases slightly when the standard deviation of $h/L$ increases from 0.001 to 0.002. A more significant increase is seen when the standard deviation of $h/L$ further increases to 0.003.

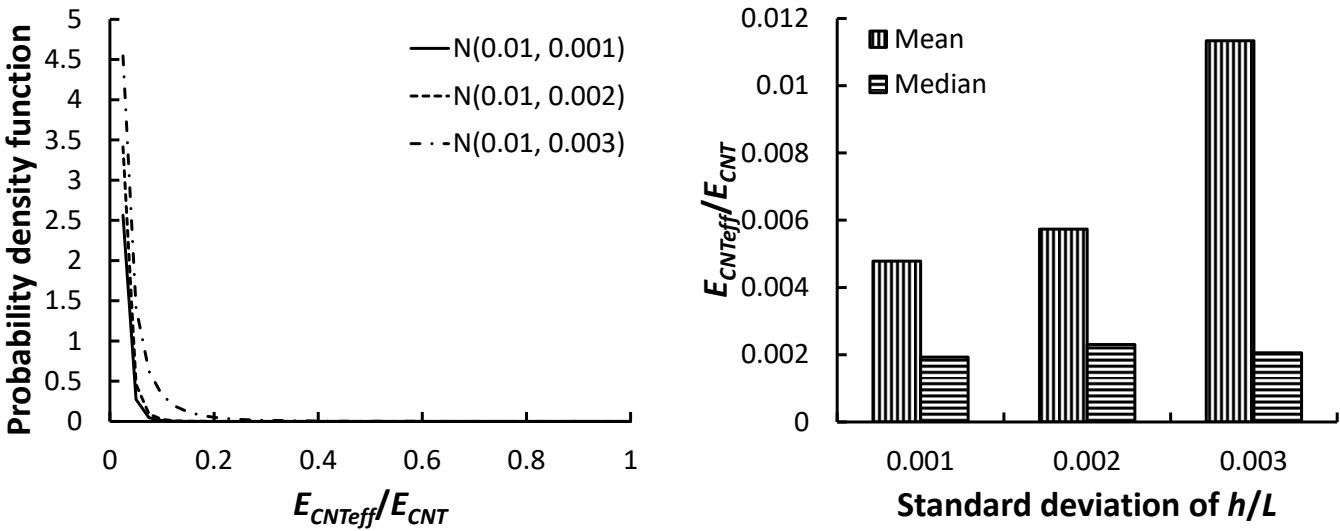

**Figure 13.** Effect of $h/L$ standard deviation on Weibull distribution for effective SWCNT moduli.

### 3.4. Effect of Waviness

To study the effect of waviness, SWCNTs of different lengths are assumed to consist of different numbers of equal curved sections of different $h/L$ ratios. The effective moduli are given in Figure 14. When $h/L$ is 0.005, the effective modulus decreases with the increase in the number of curved sections when the SWCNT length is 50 nm or 100 nm. When the SWCNT length is 200 nm, the maximum effective modulus occurs when the number of curved sections is 2. When the SWCNT length is 500 nm, the maximum effective modulus occurs when the number of curved sections is 5. When $h/L$ is 0.01, the effective modulus decreases with the increase in the number of curved sections when the SWCNT length is

50 nm. When the SWCNT length is 100 nm, the maximum effective modulus occurs when the number of curved sections is 2. When the SWCNT length is 200 nm, the maximum effective modulus occurs when the number of curved sections is 3. When the SWCNT length is 500 nm, the effective modulus increases with the increase in the number of curved sections. When $h/L$ is 0.02, the maximum effective modulus occurs when the number of curved sections is 2 when the SWCNT length is 50 nm. When the SWCNT length is 100 nm, the maximum effective modulus occurs when the number of curved sections is 3. When the SWCNT length is 200 nm, the maximum effective modulus occurs when the number of curved sections is 7. When the SWCNT length is 500 nm, the effective modulus increases with increasing number of curved sections. When $h/L$ is 0.05, the maximum effective modulus occurs when the number of curved sections is 5 when the SWCNT length is 50 nm. For the other SWCNT lengths, the effective modulus increases with the increase in the number of curved sections.

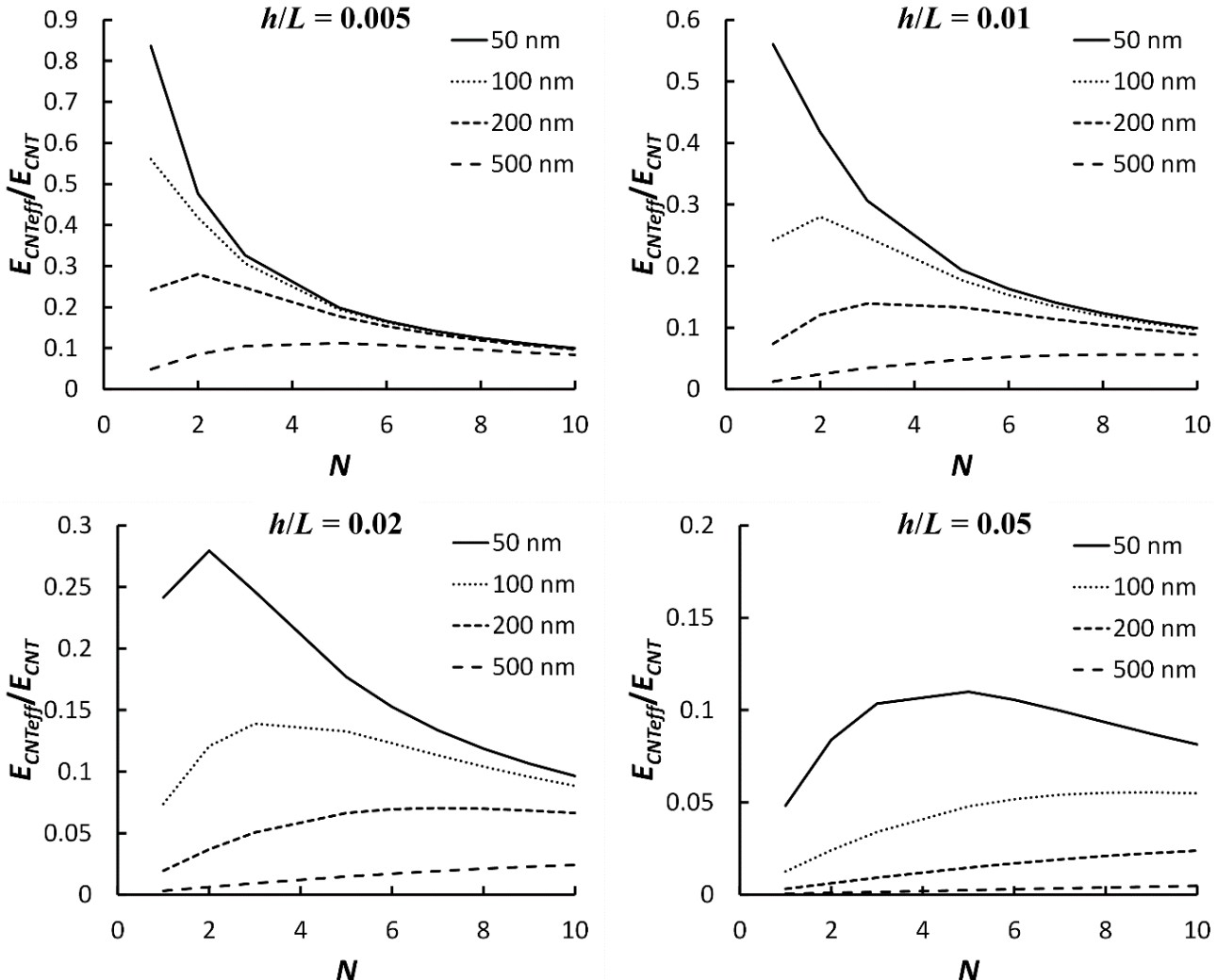

**Figure 14.** Effective modulus vs. number of curved SWCNT sections.

For evaluating the practical scenario, randomly generated section lengths are used to calculate the effective modulus of SWCNTs, and the results show the effective modulus when waviness is present is typically between 150 and 250 MPa, which is similar to the modulus of carbon fibre.

## 4. Conclusions

A simple method for determining the effective elastic modulus of curved and wavy SWCNTs is presented in this paper. The effective modulus of curved SWCNTs is derived using Castigliano's theorem. It is shown that the effective modulus decreases with increasing curvature and the length of SWCNT. For longer SWCNTs, a small increase in the curvature results in a sharp decrease in the effective modulus. The SWCNT reinforced epoxy composites are modelled with the aid of the FEA, and it is seen that the effective moduli from the FEA are in good agreement with those calculated using the presented method. Monte Carlo simulations are run with $h/L$ ratio, and SWCNT length follow various distributions. It is shown that the resulting effective moduli of SWCNTs follow Weibull distribution. The mean of the effective SWCNT moduli depends on the means of $h/L$ and SWCNT lengths, not the actual distribution type. The effective modulus of a general wavy SWCNT can be conveniently derived by considering the SWCNT as a number of curved SWCNT sections. The effective modulus when waviness is present is similar to the modulus of carbon fibre. It should be noted that only 2D curvature is considered in this study. The SWCNTs with 3D twist should be addressed in future studies.

**Funding:** This research received no external funding.

**Data Availability Statement:** All data generated or analysed during this study are included in this published article.

**Acknowledgments:** Not applicable.

**Conflicts of Interest:** The authors declare no conflict of interest.

## Appendix A

**Table A1.** Shape parameters, scale parameters, means and medians from Monte Carlo simulations.

| Set | Simulation | $h/L$ | SWCNT Length (nm) | $\beta$ | $\eta$ | $E_{CNTeff}/E_{CNT}$ | |
|---|---|---|---|---|---|---|---|
| | | | | | | **Mean** | **Median** |
| 1 | 1 | $U(0, 0.0025)$ | $N(500, 100)$ | 1.77307 | 0.55544 | 0.49432 | 0.45171 |
| | 2 | $U(0, 0.005)$ | $N(500, 100)$ | 0.92055 | 0.27978 | 0.29086 | 0.18789 |
| | 3 | $U(0, 0.01)$ | $N(500, 100)$ | 0.56502 | 0.07486 | 0.12238 | 0.03913 |
| | 4 | $U(0, 0.02)$ | $N(500, 100)$ | 0.31703 | 0.01327 | 0.09695 | 0.00418 |
| | 5 | $U(0, 0.01)$ | $N(1000, 100)$ | 0.35652 | 0.01508 | 0.07119 | 0.00539 |
| | 6 | $U(0, 0.01)$ | $N(300, 100)$ | 0.75143 | 0.24804 | 0.29487 | 0.15230 |
| | 7 | $U(0, 0.01)$ | $N(1000, 50)$ | 0.34914 | 0.01340 | 0.06799 | 0.00469 |
| | 8 | $U(0, 0.01)$ | $N(1000, 200)$ | 0.35343 | 0.01524 | 0.07411 | 0.00540 |
| 2 | 1 | $N(0.005, 0.001)$ | $N(500, 100)$ | 1.40284 | 0.05015 | 0.04570 | 0.03862 |
| | 2 | $N(0.01, 0.001)$ | $N(500, 100)$ | 0.63766 | 0.00343 | 0.00478 | 0.00193 |
| | 3 | $N(0.02, 0.001)$ | $N(500, 100)$ | 1.00000 | 0.00180 | 0.00180 | 0.00125 |
| | 4 | $N(0.01, 0.002)$ | $N(500, 100)$ | 0.63626 | 0.00410 | 0.00574 | 0.00230 |
| | 5 | $N(0.01, 0.003)$ | $N(500, 100)$ | 0.45245 | 0.00463 | 0.01134 | 0.00206 |
| | 6 | $N(0.01, 0.001)$ | $N(300, 100)$ | 0.93528 | 0.03499 | 0.03609 | 0.02365 |
| | 7 | $N(0.01, 0.001)$ | $N(1000, 100)$ | 1.00000 | 0.00033 | 0.00033 | 0.00023 |
| | 8 | $N(0.01, 0.001)$ | $N(1000, 50)$ | 1.00000 | 0.00033 | 0.00033 | 0.00023 |
| | 9 | $N(0.01, 0.001)$ | $N(1000, 200)$ | 1.00000 | 0.00033 | 0.00033 | 0.00023 |
| 3 | 1 | $U(0, 0.0025)$ | $U(300, 700)$ | 1.7050 | 0.5424 | 0.48384 | 0.43747 |
| | 2 | $U(0, 0.005)$ | $U(300, 700)$ | 0.8992 | 0.2718 | 0.28611 | 0.18081 |
| | 3 | $U(0, 0.01)$ | $U(300, 700)$ | 0.5054 | 0.0827 | 0.16220 | 0.04006 |
| | 4 | $U(0, 0.02)$ | $U(300, 700)$ | 0.3146 | 0.0124 | 0.09352 | 0.00387 |
| | 5 | $U(0, 0.01)$ | $U(100, 500)$ | 0.7780 | 0.2521 | 0.29160 | 0.15742 |
| | 6 | $U(0, 0.01)$ | $U(800, 1200)$ | 0.3289 | 0.0131 | 0.08298 | 0.00431 |

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
