# Peer review of "Effective Elastic Modulus of Wavy Single-Wall Carbon Nanotubes"

_carbon, 2023_

Round 1

Reviewer 1 Report

In the present study, the effective elastic modulus of carbon nanotubes was derived by using a new simple method, and fully compared and verified. I think this research is interesting, but some early research literature on carbon nanotubes has not been fully reviewed. For example, C. Li, C.X. Zhu, C.W. Lim, S. Li. Nonlinear in-plane thermal buckling of rotationally restrained functionally graded carbon nanotube reinforced composite shallow arches under uniform radial loading, Applied Mathematics and Mechanics-English Edition, 2022, 43(12): 1821–1840. The author should pay attention to these papers in the revised version to expand the scope of literature review.

Author Response

Thanks for the comments.  As suggested, the literature review has been expanded in the revised manuscript.  The suggested paper has been included.

Reviewer 2 Report

I believe the manuscript matched the scientific scope of C Journal of Carbon Research, and can be considered for publication without corrections.

Author Response

This reviewer is very supportive to this paper and no corrections are needed.  Thanks.

Reviewer 3 Report

The manuscript is devoted to a method for determining the effective elastic modulus of curved and wavy SWCNTs. The paper has interesting potentialities, which are currently not fully developed. Some points need to be carefully checked. A deep reorganization seems needed. 

 Please, try to avoid repetitions: lines 28-29 ``... and make them less useful for certain applications....'' and lines 34-35 ``... , making them less useful for certain applications...''. Similarly, in line 40: ``...more than one type of inclusion type...''

The sentence ``The effective modulus of a general wavy SWCNT is derived by considering the SWCNT as a number of curved SWCNT sections'' is not very clear.

While correct, Eqs. (1) and (2) are introduced, but not used

Please, check Eq. (5). The exponent 2 for the power is lacking.

The estimate of the shear coefficient is lacking. It depends on many aspects of the cross-section. The Authors should clarify this aspect. In addition, for slender beams, the shear contribution is usually much smaller with respect to the others. Also, the contribution of axial force may be negligible with respect to that of the bending moment, depending on the ratio between eccentricity and gyration radius. Did the Authors consider these aspects?

Please, check Eq. (8). I could not obtain Eq. 8, from the previous equations. For instance, I obtain \delta_{tension}=2PR (pi/2-\theta_0)(sin(\theta+\theta_0))^2/EA. Maybe I made mistakes, but it is better to check. 

How Eq. (9) is computed? Similarly to Eq. (10)?

Sect. 2.2 does not contain relevant details on the FEA.

Sect. 2.3 does not contain relevant details on the Monte Carlo Simulation.

What does the figure at the beginning of page 6 represent? Apparently, it is not numbered and its caption is lacking. 

Reviewer 4 Report

The authors derived an approximate expression of the elastic modulus of single-walled carbon nanotubes (SWNTs) with arched geometry. The validity of the derived approximation, based on Castigliano's method, was verified by comparison with finite element analysis. Although the results presented in this manuscript can be evaluated to have some degree of novelty, I believe that the following points should be reconsidered before it is accepted for publication.

[1] The title presented is misleading. I think it is necessary to include words such as "wavy" and "curved" in the title in order to correctly convey the purpose of the paper to the readers. Otherwise, the reader will mistakenly think that this paper deals with the elasticity of ordinary straight SWNTs, which is an old-fashioned problem.

[2] The "wavy"-shape that was assumed by the author in this manuscript is limited to the shape of an arc drawn on a two-dimensional plane, as demonstrated in Figure 1. In other words, curves with twists in three-dimensional space are out of scope. However, the wavy SWNTs that are actually synthesized should be more of the latter with twists. This point as well as the limitation of the present theory should be commented in the manuscript.

[3] In Line 64, it is mentioned that the outer diameter of SWNTs is taken to be 1.3 nm. But why? This value is by no means a typical SWNT outer diameter.

[4] Moreover, the derivations that follow Line 64 do not require a fixed value of the outer diameter. Actually in Line 84, the cross section of SWNT is expressed by a variable A, not a specific value derived from the outer diameter being 1.3 nm.

[5] The definitions of the variables Ef and Gf given in Equation 8 are not written in the manuscript.

[6] Figure 4 illustrates that the effective elastic modulus decreases significantly as the length of the tube increases, even though the degree of waviness indicated by the value of h/L remains unchanged. However, there is no explanation as to why such a remarkable change occurs. An explanation from a physical or structural mechanics viewpoint must be required.

Round 2

Reviewer 3 Report

The revised version of the paper contains many improvements. The Author replied to my comments in a correct and satisfactory way. I have no further questions, comments, or suggestions. In my opinion, the paper is worthy of publication as is.

Author Response

I am glad that this reviewer is happy about the revision that I have made.  I would like to thank their valuable suggestions.

Reviewer 4 Report

After reading the revised manuscript, I find it very regrettable that the author did not respond to my questions/comments very sincerely. In particular, regarding my previous comments [3] and [4], I do not understand at all what the author's intention is in her/his responses. If a manuscript that ignores the comments of reviewers is accepted in this way, I feel that this journal has lost its academic impartiality.

[3]
The author might have thought that simply inserting the word "wavy" in a few places in the text would have addressed this problem. But to solve this problem, the text should elaborate on what the author means by the word "wavy".

[4]
The response from the author does not answer my question at all. Even if the diameters of SWNTs that are actually synthesized are in the range of 0.8 nm to 2.4 nm, this should not be a reason to consider SWNTs with a diameter of only 1.3 nm in this paper. I have strong doubts about the author's unwillingness to answer my question with complete sincerity

Author Response

Thanks for the comments.

[3] I have added more explanations about the waviness of SWCNT.  See page 2.

[4] I have added more explanations and references about the diameters of SWCNT, based on which 1.3 nm was chosen.  See page 4.

Thank you.

Round 3

Reviewer 4 Report

n/a